# “*It Soothes Your Heart*”: A Multimethod Study Exploring Acceptability of Point-of-Care Viral Load Testing among Ugandan Pregnant and Postpartum Women Living with HIV

**DOI:** 10.3390/diagnostics14010072

**Published:** 2023-12-28

**Authors:** Agnes Nakyanzi, Faith Naddunga, Michelle A. Bulterys, Andrew Mujugira, Monique A. Wyatt, Brenda Kamusiime, Alisaati Nalumansi, Vicent Kasiita, Sue Peacock, Connie L. Celum, Norma C. Ware

**Affiliations:** 1Infectious Diseases Institute, Makerere University, Kampala P.O. Box 7072, Uganda; fnaddunga@idi.co.ug (F.N.); amujugira@idi.co.ug (A.M.); bkamusiime@idi.co.ug (B.K.); analumansi@idi.co.ug (A.N.); vkasiita@idi.co.ug (V.K.); 2Department of Global Health, University of Washington, Seattle, WA 98109, USA; mbult@uw.edu (M.A.B.); peacocks@uw.edu (S.P.); ccelum@uw.edu (C.L.C.); 3Department of Global Health and Social Medicine, Harvard Medical School, Boston, MA 02115, USA; monique_wyatt@hms.harvard.edu (M.A.W.); norma_ware@hms.harvard.edu (N.C.W.); 4Harvard Global, Cambridge, MA 02138, USA; 5Departments of Medicine and Epidemiology, University of Washington, Seattle, WA 98109, USA; 6Department of Medicine, Brigham and Women’s Hospital, Boston, MA 02115, USA

**Keywords:** point-of-care viral load testing, POC VL, viral load, ART adherence, PMTCT

## Abstract

Background: High adherence to antiretroviral therapy (ART) is critical for achieving viral suppression and preventing onward HIV transmission. ART continuation can be challenging for pregnant women living with HIV (PWLHIV), which has critical implications for risk of vertical HIV transmission. Point-of-care viral load (POC VL) testing has been associated with improved treatment and retention outcomes. We sought to explore acceptability of POC VL testing among Ugandan PWLHIV during pregnancy and postpartum. Methods: This multimethod analysis drew on quantitative and qualitative data collected between February and December 2021. Quantitatively, we used an intent-to-treat analysis to assess whether randomization to clinic-based POC VL testing during pregnancy and infant testing at delivery was associated with improved viral suppression (≤50 copies/mL) by 3 months postpartum compared to standard-of-care (SOC) VL testing through a central laboratory, adjusting for factorial randomization for the male partner testing strategy. Additionally, a subset of 22 PWLHIV in the POC VL arm participated in in-depth qualitative interviews. We inductively analyzed transcripts to develop categories representing concepts that characterized women’s perceptions of POC VL testing during pregnancy and at delivery and ways that POC VL testing may have impacted their ART adherence and viral suppression. Key themes around women’s perceptions of POC VL testing were then organized into main categories. Results: Overall, 151 PWLHIV were enrolled into the study, 77 (51%) of whom were randomized to receive POC VL testing during pregnancy and at delivery. Women reported in qualitative interviews that POC VL testing had (1) motivated their ART adherence during pregnancy and postpartum and that they felt this testing method had (2) helped them protect their infants from acquiring HIV and (3) improved their emotional wellbeing. Conclusions: POC VL testing was highly acceptable among Ugandan PWLHIV and was viewed as an important tool that women believed improved their ART adherence, gave them information necessary to protect their infants from vertical HIV acquisition, and improved their emotional wellbeing. These findings support the global scale-up of POC VL testing in settings with high HIV burden, especially for PWLHIV who may be at risk of treatment disruptions or loss to follow-up.

## 1. Background

Daily adherence to antiretroviral therapy (ART) is critical for achieving viral suppression, which is defined by the World Health Organization as having a viral load (VL) < 1000 copies/mL [1,2,3]. Maintaining viral suppression is essential for improving health, reducing the risk of adverse outcomes or opportunistic infections and preventing onward transmission [1]. Viral load is the most important indicator of ART adherence, and maternal VL during pregnancy and postpartum is the most predictive factor for vertical HIV transmission risk [4,5]. However, remaining on ART therapy and achieving viral suppression has been shown to be especially challenging for pregnant women living with HIV (PWLHIV) [1,6,7,8]. PWLHIV face individual and societal barriers to ART adherence, such as anticipated and experienced stigma, fear of side effects, unwanted HIV disclosure to their partner and family, poor mental health, low health literacy, and intimate partner violence [6,9,10]. Suboptimal retention rates have been observed among PWLHIV in Uganda in prevention of vertical transmission programs, including high loss to follow-up (>30%) in the first year post-ART initiation, particularly among women who did not receive regular VL testing [11,12]. Given the implications this might have for infant HIV acquisition, especially within the first three months postpartum, person-centered interventions are urgently needed to improve ART adherence among PWLHIV.

The WHO currently recommends that all people living with HIV receive VL testing at least every 6 months [5,13]. Current standard-of-care viral load (SOC VL) testing involves sending blood samples to a central laboratory for processing, and PWLHIV often will not receive their viral load results and counseling until their next clinical visit every 1–3 months or quarterly. Point-of-care viral load (POC VL) testing offers an improved and effective option for same-day VL testing and has been significantly associated with improved retention in care [14,15]. Based on a strong body of evidence around improved ART retention with POC VL testing, the WHO issued updated guidance in 2021 supporting the implementation of POC VL testing with a special focus on reaching PWLHIV [16]. PWLHIV are a vulnerable population at higher risk of poor ART adherence, which has implications for vertical HIV transmission risk to infants. To tackle this issue, the Ugandan Ministry of Health updated their national HIV guidelines in 2023 to now recommend that all PWLHIV receive VL testing every 3 months during pregnancy and breastfeeding. This multimethod study sought to explore acceptability of POC VL testing among Ugandan PWLHIV alone or combined with an intervention to encourage their male partners to test for HIV and mutually disclose their HIV status as part of a multipronged approach to support ART use and viral suppression among PWLHIV.

## 2. Methods

### 2.1. Study Design and Population

The Kingasa study (Kingasa is Lugandan for “It benefits me”; Clinicaltrials.gov: NCT05092997) was a pilot randomized trial that enrolled PWLHIV recruited from a public antenatal clinic in Kampala, Uganda, who were at least 18 years of age and had partners of unknown HIV status between February and December 2021. This study assessed the role of POC VL testing and same-day adherence counseling as well as interventions to improve male partner engagement on women’s motivation to adhere to ART and achieve viral suppression throughout pregnancy and 3 months postpartum. Women were randomized 1:1:1:1 to four randomization groups using a factorial design: (1) POC VL testing with same-day ART adherence counseling + SOC invitation letter for male partners to test for HIV; (2) POC VL testing with same-day adherence counseling + intervention invitation letter for male partners for a comprehensive wellness visit, including rapid dual HIV and syphilis testing, blood pressure monitoring, and visual acuity screening; (3) SOC VL testing with next-visit ART adherence counseling + SOC invitation letter; and (4) SOC VL testing with next-visit ART adherence counseling + intervention invitation letter for the wellness visit. For the present analysis, we grouped all mothers who were randomized to POC VL testing vs. SOC VL testing, regardless of the male partner intervention arm to which they were also randomized. All multivariate analyses assessing differences in outcomes by VL testing method were also adjusted for the male partner intervention randomization arm. HIV testing for infants was conducted as study procedures per national guidelines using standard dried blood spot samples taken at 6 weeks of age. Partway through the study, a subset of mothers who were randomized to the POC VL arm additionally received POC VL testing with infant HIV diagnostic testing for early infant diagnosis (EID) at the time of delivery to assess feasibility and acceptability. SOC VL testing was performed according to national guidelines six months following ART initiation and every 12 months thereafter if virally suppressed. For this study, the SOC VL group provided their blood samples for two time points of viral load testing: (1) at their enrollment visit during pregnancy, with the results and counseling received 3 months later at their next study visit, and (2) at their visit 3 months postpartum, with the results received at the clinic 3 months later (6 months postpartum, Figure 1).

POC VL testing was performed using the GeneXpert^®^ HIV-1 viral load test (Cepheid, Sunnyvale, CA, USA) [17], which has been tested and validated for use in sub-Saharan African settings with a sensitivity and specificity of 94% and 99%, respectively [18]. This self-contained, rapid polymerase chain reaction (PCR) test detects the number of HIV copies per milliliter of plasma in an average time of 90 min. The POC VL machine used in this study could process up to four samples simultaneously, allowing for larger testing volumes for optimized efficiency. At study visits with VL testing for women in the POC VL arm, research nurses collected a blood sample at the beginning of the clinical encounter so that the laboratory technician could process it. At the same time, participants completed their study visit. As soon as the laboratory technician processed the POC VL results, the research nurse delivered the results to the mother, accompanied by personalized VL and adherence counseling. When samples were collected late in the afternoon close to laboratory closure, results and counseling were provided to mothers in the POC VL arm the following day via telephone.

### 2.2. Data Collection

PWLHIV were followed every three months until they reached three months postpartum; VL testing was not conducted at every study visit. Socio-demographic characteristics were collected at baseline, and birth outcomes were collected at follow-up. Throughout pregnancy, at delivery, and at the final study visit (3 months postpartum), PWLHIV received either SOC or POC VL testing based on their study arm (Figure 1). All participants were asked at their visit 3 months postpartum to self-report their perceived ART adherence in the past 30 days using a 5-point Likert scale (excellent, very good, good, fair, poor, and very poor). A purposively sampled subset of PWLHIV were invited to participate in a single in-depth qualitative interview; only PWLHIV who were randomized to the POC VL arm and had already received POC VL testing were recruited. The selection criteria for the qualitative subsample were PWLHIV who (a) had received POC VL at the time of delivery, (b) had not yet completed their exit visit at 3 months postpartum, and (c) were reachable by telephone for scheduling. Interviews were conducted by trained Ugandan qualitative researchers and lasted approximately 60 min each. A comprehensive interview guide explored women’s perceptions around receiving POC VL testing during pregnancy and at delivery and their experiences receiving test results and counseling on the same day or the following morning if the sample was provided late in the day prior. Interviews were audio recorded with permission and were transcribed and translated verbatim from Luganda to English by the interviewers.

### 2.3. Data Analysis

This analysis drew on both quantitative and qualitative data and only includes data for women. Quantitatively, we evaluated the primary outcome of a proportion of women who were virally suppressed (defined in this study as ≤50 copies/mL) at 3 months postpartum by randomization arm (POC VL testing compared to SOC VL testing) using intention-to-treat analysis and adjusting for additional randomization for male partner testing. Quantitative data were analyzed descriptively using R programming (version 09.2, www.R-Project.org, Vienna, Austria, accessed 1 October 2023). Descriptive statistics were used to characterize the overall sample of PWLHIV at baseline, and multivariate log binomial regression models with robust standard errors were used to estimate the prevalence ratio of achieving viral suppression. Qualitative analysis consisted of inductively analyzing transcripts to develop categories representing concepts that characterized women’s perceptions of POC VL testing during pregnancy and postpartum and ways that POC VL testing may have encouraged their ART adherence. Transcripts were analyzed by authors A.N., F.N., and M.A.B. to identify key concepts related to POC VL testing. A total of 20% of transcripts were analyzed by two analyzers separately for consistency; concept development and discrepancies were discussed as a team with guidance from N.C.W., M.A.W., C.C., and A.M. Key themes around women’s perceptions of POC VL testing were then organized into main categories.

### 2.4. Ethical Approvals

This study was approved by the University of Washington Human Subjects Review Committee (STUDY00009286), the Mildmay Uganda Research Ethics Committee (MUREC 0707-2020), and the Uganda National Council for Science and Technology (HS991ES). All participants provided written informed consent.

## 3. Results

### 3.1. Study Participant Characteristics

Of the 151 PWLHIV enrolled into the Kingasa study, 77 (51%) were randomized to receive POC VL testing throughout their pregnancy and 74 (49%) were randomized to receive SOC VL testing (Table 1). Among all women, 25% received seven years or less of formal schooling, and 48% were employed. Women’s median age was 28 years (interquartile range (IQR): 24–32), and the majority (92%) of women were married. At study enrollment, 17% of women had been diagnosed with HIV in the prior 2 months, 20% were diagnosed 3–24 months prior, and 64% were diagnosed >24 months prior to enrollment. At enrollment, almost all women were already taking ART, 83% had an undetectable viral load, and 40% had disclosed their HIV status to their male partners.

### 3.2. Pregnancy and Treatment Outcomes

All but nine pregnancies (7%) resulted in live births, of which all born babies were confirmed HIV negative (Table 2). When asked to self-report their ART adherence, 66% rated their adherence as excellent, 31% rated it good/very good, and 3% rated it fair/poor. Overall, 86% had an undetectable viral load at 3 months postpartum. There was no significant difference in the proportion of women who were virally suppressed between women randomized to the POC VL testing arm and women randomized to the SOC VL testing arm (88% vs. 81%, Table 2; prevalence ratio (PR) 0.52, 95% confidence interval (CI): 0.17, 2.02, *p* = 0.40). The randomization for male partner testing interventions was not associated with any differences in postpartum viral suppression or self-reported ART adherence.

### 3.3. Qualitative Sample Characteristics

A total of 22 PWLHIV enrolled in Kingasa (15%) participated in in-depth interviews. The median age of women captured in the qualitative sample was 27.5 years (IQR: 25–33.5), and 5 (23%) had not completed primary school (7 years of education). All but two women (91%) were currently married. The sociodemographic characteristics were comparable between the overall sample of women and the subset of women included in the qualitative sample.

### 3.4. Exploring Acceptability of POC VL Testing among Ugandan PWLHIV

Overall, PWLHIV participating in this qualitative study reported that POC VL testing had (1) motivated their ART adherence during pregnancy and postpartum and that they felt this testing method had (2) helped them protect their infants from acquiring HIV and (3) improved their emotional wellbeing. These categories are organized and presented below in this order (Figure 2).

#### 3.4.1. Category 1: POC VL Testing Encourages Maternal ART Adherence and Viral Suppression

In interviews, women described the importance of ART adherence in order to achieve viral suppression and explained that a low viral load would keep them healthy, whereas a high viral load could lead to poor health outcomes, even death. As a result, women in this study referred to POC VL testing as “*lifesaving*” because it encouraged their ART adherence (“*I prefer same day results because I can base on the results to save my life*”). POC VL testing was also viewed as an “upgrade” from SOC VL testing as it no longer meant they had to wait until their next clinical encounter (quarterly), which could mean waiting for up to several months to receive their VL results. Women also described how receiving POC VL results was highly motivating for their ART adherence regardless of the direction of the results. Women reported that receiving VL results indicating that they were not virally suppressed motivated them to improve their ART adherence and offered them an opportunity to make immediate changes in their ART adherence.

“*[After receiving my POC VL results], I reacted and decided to take ART like no man’s business, because I got scared that if I continue being careless, I may die soon. When you are told that your virus is not suppressed, you know what to do when you go back home. If you have not been adhering well to your drugs, you have to change immediately. This is not the case when you leave hospital without knowing your results. You cannot know if you are not adhering well to your drugs until when you go back to your next visit.*” PWLHIV, 25 years old.

Results indicating viral suppression were viewed as positive reinforcement, confirming that they had been taking their ART treatment well and motivating them to continue. They described the joy associated with receiving POC VL results confirming that they were virally suppressed, which made them feel as if they had permanently “*cured*” themselves of HIV. Even though only a few women joked about this, the perception that viral suppression was just as good as curing HIV served as a strong motivator to continue adhering well to their antiretroviral medications and to keep their viral load suppressed.

“*[After receiving my POC VL results], I felt good and happy because it felt like as if I had cured [myself] of HIV (she laughs). I was strong because I knew that I was adhering well to my drugs and there was no way the results would show otherwise. I did not have any fears at all.*” PWLHIV, 30 years old.

In addition to providing important information about ART adherence, mothers also felt that POC VL testing prompted necessary clinical action when needed to improve their health. For example, a few women who had been adhering well to their medication but were still struggling to achieve viral suppression described how POC VL testing prompted their health workers to quickly prescribe them with a different ART regimen.

“*[Without POC testing] I think we could have died. Can you imagine living without knowing what is happening in your body? The viral load test is very important because it helps you to know whether you are adhering well to your drugs or not and also it helps the health workers to know if the drugs they are giving you are working well. For example, when I did my third viral load test, I was told that my virus was not suppressed and the health worker decided to change the regimen for me. If the test was not done, the health worker would not have known that the drugs are no longer working well for me.*” PWLHIV, 27 years old.

#### 3.4.2. Category 2: POC VL Testing Helps Mothers Protect Their Babies from Acquiring HIV

Mothers expressed strong motivation and pressure to keep their babies free from acquiring HIV (“*I did not want that for my baby, I can fight as an adult but my little one might not know how to fight*”). They reported that POC VL results gave them the knowledge and information necessary to make rapid decisions that could impact their babies’ health, particularly concerning the prevention of vertical HIV acquisition. Women believed that POC VL testing gave them the control and agency to reduce this risk during pregnancy, at delivery, and during breastfeeding. During pregnancy, POC VL results served as a reminder and catalyst for mothers to achieve viral suppression prior to delivery (“*learning the results on the same day is best for me because I get to know if I can infect the child or not when I am pregnant”*). At the time of delivery, POC VL testing was a helpful indicator of the risk of vertical transmission during labor, which was desirable for both the mother and health worker to know. Moreover, mothers described the immense joy of having an HIV-negative baby (“*there is nothing as sweet as that")*, further empowering them to continue adhering well to their drugs.

“*I got to know about my high viral load results four months before delivery and I fought to suppress it. If I had taken the other one [SOC testing], I would have got results when it is one month for me to deliver so I would have delivered with a high viral load which puts the baby on high risk of getting HIV. I wouldn’t have known and I wouldn’t have reacted the way I reacted. The other thing is the joy of having an HIV negative child. I was scared to miss that joy because I brought my first baby on earth when is negative and I was happy so I thought about that joy and I said I should take ART to avoid transmitting HIV to my unborn baby.*” PWLHIV, 25 years old.

“*I am sure the effort I put in taking ART helped me so much to give birth to a negative baby because you never know, I would have given birth to an infected baby if my viral load was not worked upon.*” PWLHIV, 23 years old.

POC VL results also directly influenced mothers’ decisions on whether they felt it was safe enough to breastfeed their babies immediately following birth. Mothers recounted asking and ensuring that the health workers were going to test their viral load immediately at delivery as having their VL results was influential in their decisions around whether to consider alternative feeding options to continue protecting their infant until they were able to achieve viral suppression.

“*Once you have the results then you can decide whether to breastfeed the child or not. It was the reason why I asked to have my viral load tested immediately I gave birth. There was no way I could put my child at a risk of breastfeeding when I know my viral load is high. The chance of the baby getting HIV would be high but now I started breastfeeding when I am comfortable that my baby will not get HIV from me since my viral load was suppressed. At that time, I was thinking about protecting my baby and that is why I asked for viral load test so that I decide well.*” PWLHIV, 25 years old.

#### 3.4.3. Category 3: POC VL Testing Improves Mothers’ Emotional Wellbeing

Women generally felt anxious about receiving their VL results, regardless of how well they felt they were adhering to their ART. The VL results signified the proof that they were performing well, but women feared that pregnancy alone could interfere with viral suppression. Additionally, women who may have missed a dose or taken a dose late were afraid that the VL results would reflect this. Women in this qualitative sample strongly preferred POC VL testing over SOC VL testing because receiving viral load results on the same day alleviated negative feelings and gave women the opportunity to take immediate actions based on the results they received without the added stress of waiting. Women perceived that POC VL testing alleviated their anxieties around VL testing, thus improving their emotional wellbeing, particularly by alleviating the anxiety and stress that came from waiting until their next visit to receive their viral load results. Women acknowledged they held an expectation of going to the hospital to understand what might be “*disturbing*” them, and not receiving feedback following medical testing immediately caused uncertainty, stress, and frustration.

“*(Respondent chuckles). [I prefer receiving my results] that same day. When you go to hospital while sick, you expect to be told what is disturbing you. When you go back without knowing, it stresses you a lot and you cannot settle without knowing what is disturbing you. It is important for one to know because it helps you to know what to do next. When you are not told whether your viral load is suppressed or not, you keep wondering what the results will be whereas if you know the results, you go back home knowing whether to continue adhering well or improve adherence if you have not been adhering well to your drugs.*” PWLHIV, 37 years old

“*I prefer the one where I get my results immediately because it soothes your heart. The viral load test where you have to get the results on your next visit is not good because you are always worried about the results that you will be given. You always wonder what the health workers will tell you when the viral load is not suppressed. You are always looking at the phone to see if maybe a health worker will call you and when he/she calls, you still wonder what the health worker will tell you. Living in fear is not good at all. I would rather get my results immediately than wait.*” PWLHIV, 28 years old

Women also described how anxieties from waiting for their VL results would negatively impact their health (“*you even get high blood pressure!*”*),* their viral load, CD4 count, and ART adherence (“*you will be worried all the time and even fail to take your pills well*”*)*. POC VL testing made women feel “*happy*” and relieved that their health would not be further impacted by the waiting.

“*[POC VL testing] makes me happy because I do not worry about that all the time… for the standard VL test, I could spend much time worried thinking: “I wonder what the results will be like? I wonder what the VL is?” and yet remember that the more you worry, the more your CD4 count lowers. Yet, for the quick test when you learn your results early, it makes you happy.*” PWLHIV, 23 years old.

## 4. Discussion

This multimethod study assessed acceptability of POC VL testing during pregnancy and postpartum among Ugandan PWLHIV. Women in this study exhibited a strong understanding of the purposes of VL testing and the role of ART adherence in achieving viral suppression. They reported that receiving their POC VL results served as confirmation they were adhering well to ART and/or as a motivator to improve their adherence. The qualitative evidence demonstrated that PWLHIV strongly preferred POC VL testing over SOC VL testing, and they believed that POC VL helped them improve their ART adherence, emotional wellbeing, and ability to protect their babies from vertical HIV acquisition during pregnancy, delivery, and breastfeeding.

Though our quantitative analysis did not find a statistically significant difference in women’s achievement of viral suppression by VL testing type, this pilot study was limited by a small sample size and limited statistical power given that >80% of PWLHIV were already virally suppressed at study enrollment. A larger study involving a greater proportion of virally unsuppressed PWLHIV at baseline is needed to assess differences in treatment outcomes by VL testing type over time. The qualitative findings captured important insights into the reasons why women believed that POC VL testing directly improved their motivation to adhere to ART compared to SOC VL testing and is likely still an important factor, even if not reflected statistically. This demonstrates the value of a multimethod approach as the qualitative analysis provided greater context beyond the quantitative findings alone.

Few qualitative studies have assessed acceptability of POC VL testing among adults living with HIV. Two qualitative studies, one from South Africa [19] and another from Kenya [12], interviewed clients and providers who noted clear practical benefits of implementing POC VL testing, namely, faster turnaround times, fewer clinical encounters, and reduced burden on clients and providers. POC VL testing among PWLHIV has been found to significantly increase ART initiation in pregnancy and reduce the risk of vertical HIV transmission [20]. Women in our study described how their POC VL results served as an important marker of how well they were adhering to their antiretroviral medications and as an indicator of the level of risk of transmitting HIV to their infants during pregnancy or breastfeeding. Additionally, the same-day, personalized counseling helped women feel more confident and in control of their own abilities to protect their infant from HIV acquisition through improved ART adherence. Women who received POC VL test results that showed they were not virally suppressed during their pregnancy viewed this as an urgent reminder to improve their ART adherence and felt they could have given birth to an infant living with HIV had they not received prepartum POC VL testing. Women also described how, after delivery, knowing their POC VL results gave them decision-making autonomy over whether to breastfeed their infant right away.

Women in our study perceived that POC VL testing helped healthcare workers make rapid decisions in their treatment management, particularly expediting necessary ART regimen changes. The STREAM trial in South Africa found that POC VL testing among adults living with HIV significantly improved viral suppression and retention in care compared to SOC VL testing [14]. This was attributable to the faster turnaround time of results and counseling and faster clinical actions taken in the event of viral failure [14,21]. Similarly, an observational cohort study found Zimbabwean PWLHIV who received POC VL testing had a significantly higher likelihood of receiving clinical action (e.g., changed ART regimen, drug resistance testing, and closer follow-up and counseling) than PWLHIV who received SOC VL testing [20].

Women reported that VL testing in general was anxiety inducing, but PWLHIV perceived that POC VL testing had a positive impact on their emotional wellbeing. Women in our study described the added pressures they felt to have children free from HIV, which led them to feeling anxious while awaiting their VL results as high viral load was indicative of higher risk of vertical transmission. Women expressed that POC VL testing eliminated the long waiting time typically experienced in SOC VL testing and, as a result, reduced anxieties associated with waiting and worrying. Therefore, not only can POC VL testing improve treatment outcomes but it also has the potential to improve emotional wellbeing among PWLHIV, a vulnerable subset of adults living with HIV.

Qualitatively, mothers described the many perceived ways in which POC VL testing improved their ART adherence. However, quantitatively, a lower proportion of mothers in the POC VL arm self-reported “excellent” ART adherence at postpartum visits compared to mothers in the SOC VL arm (60% vs. 73%, Table 2). This discrepancy between the quantitative and qualitative data may be partly explained by mothers in the POC VL arm refraining from overselling or overestimating their ART adherence to the healthcare worker because, due to POC VL testing, their VL and therefore their adherence would be known and discussed at the same clinical encounter. This would not be as great a concern for mothers in the SOC VL testing arm, who would only receive their VL results months later at their next visit. Mothers in the POC VL arm often described how important it was that they made their healthcare workers proud of them based on their achievement of viral suppression. Additional hypotheses for reasons behind this discrepancy should be explored further in future studies.

As HIV treatment initiation and monitoring guidelines recommend less CD4 testing, the demand for decentralized VL testing as the preferred method for HIV treatment monitoring is growing substantially across sub-Saharan Africa [22]. With centralized laboratories overburdened by processing needs for HIV, TB, COVID-19, and other diseases, POC VL testing is the preferred method for scale-up in high-burden HIV settings thanks to reduced turnaround times. Reassuringly, POC VL testing in sub-Saharan Africa has shown that it can be accurately and feasibly implemented at reasonable costs [23,24,25]. For example, the average costs of a POC VL test and a SOC VL test using centralized laboratory testing were USD33.71 and USD28.62, respectively, in Malawi, and USD24.25 per POC VL test and USD25.65 per SOC VL test in Kenya [23,24]. POC VL testing implementation would require a dedicated and trained POC technician, which may require task shifting among staff. POC VL testing can be accurately conducted by trained nonlaboratory, lay staff, which may be necessary in more rural, resource-limited clinics or communities [26]. Innovative and cross-cutting solutions are needed to address common health system challenges with POC VL testing, including equipment procurement and maintenance, supply chain management for cartridges, and limited staff availability for after-hours/weekend processing [25].

Our study’s strengths included the collection of both quantitative and qualitative data from a randomized controlled trial comparing treatment outcomes of POC VL with SOC VL testing among PWLHIV. The inclusion of qualitative data provided a view of the impact of POC VL testing on women’s clinical outcomes that were not necessarily accessible through quantitative inquiry alone. Additionally, this study explored acceptability of POC VL testing during pregnancy and breastfeeding from the perspectives of PWLHIV instead of implementers. Limitations included pending viral load results at study follow-up, particularly in the SOC VL arm, and potential sampling biases. This study sampled PWLHIV aged ≥18 years with male partners of unknown HIV status attending public antenatal care clinics and may not be fully representative of all PWLHIV in Uganda who could benefit from POC VL testing. One outcome of this analysis, self-reported ART adherence at the postpartum visit as measured by the Likert scale, was not collected at baseline, thus limiting our ability to assess changes in ART adherence between the time of baseline and the visit 3 months postpartum by randomization arm. Given the short follow-up time and small sample size for this pilot RCT, we could not assess changes in ART adherence over time. This study was conducted in a controlled research setting with in-depth, personalized counselling from study nurses. Therefore, the findings may not be reflective of POC VL implementation in real-world settings.

## 5. Conclusions

POC VL testing was highly acceptable among this study population of Ugandan PWLHIV. Qualitative findings indicated that women viewed POC VL testing as an important tool that improved their ART adherence, gave them information and motivation to protect their children from vertical HIV acquisition during pregnancy and postpartum, and improved their emotional wellbeing. These findings support additional evaluation of POC VL testing in settings with high HIV burden, especially for PWLHIV who have detectable VL in their initial antenatal visit or report issues with ART adherence.

## Figures and Tables

**Figure 1 diagnostics-14-00072-f001:**
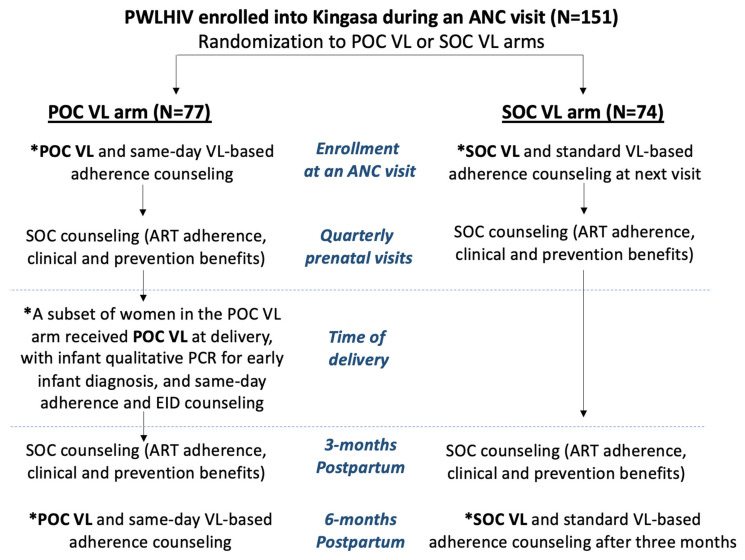
Flowchart of study design and procedures by randomization arm. Legend abbreviations: PWLHIV = pregnant women living with HIV; POC VL = point-of-care viral load; SOC VL = standard-of-care viral load; PCR for EID = polymerase chain reaction for early infant diagnosis; VL-based counseling = viral-load-based counseling. * Visits that involved blood sample collection for viral load testing.

**Figure 2 diagnostics-14-00072-f002:**
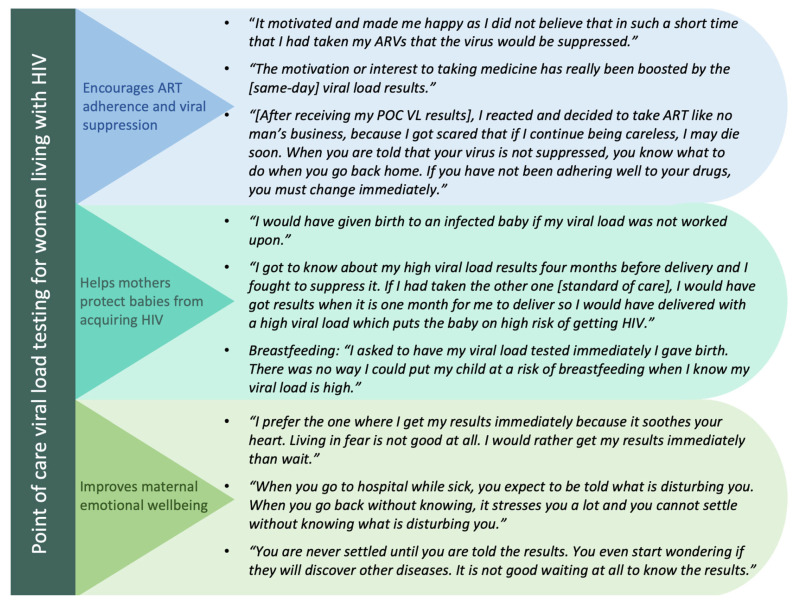
Summary of main categories, accompanied by illustrative quotes.

**Table 1 diagnostics-14-00072-t001:** Population characteristics *n* (%) and median (IQR).

Characteristic	POC VL, N = 77	SOC VL, N = 74	Overall, N = 151
Employed	38 (49%)	34 (46%)	72 (48%)
Education <7 years	20 (26%)	17 (23%)	37 (25%)
Age (years)	28.0 (25.0, 32.0)	27.0 (24.0, 31.0)	28.0 (24.0, 32.0)
Marital status			
Married	70 (91%)	69 (93%)	139 (92%)
Never married	5 (6.5%)	5 (6.8%)	10 (6.6%)
Divorced/separated	2 (2.6%)	0 (0%)	2 (1.3%)
Relationship is polygamous	28 (36%)	23 (31%)	51 (34%)
Number of previous live births			
None	12 (20%)	13 (25%)	25 (23%)
1–2	15 (25%)	16 (31%)	31 (28%)
3+	32 (54%)	23 (44%)	55 (50%)
Knew she was living with HIV when she started this pregnancy	55 (72%)	57 (77%)	112 (75%)
Time since HIV diagnosis (months)			
≤2 months	12 (16%)	13 (18%)	25 (17%)
3–24 months	16 (22%)	13 (18%)	29 (20%)
25+ months	46 (62%)	48 (65%)	94 (64%)
On ART at enrollment	75 (99%)	72 (97%)	147 (98%)
Number of days lapsed since last ART dose			
0 days (<24 h)	45/75 (60%)	41/73 (56%)	86/148 (58%)
1 day (24–48 h)	30/75 (40%)	30/73 (41%)	60/148 (41%)
2+ days	0/75 (0%)	2/73 (3%)	2/148 (1%)
Undetectable viral load (<50 copies/mL) at enrollment	59 (86%)	56 (80%)	115 (83%)
Disclosed HIV status to partner by enrollment	35 (45%)	25 (34%)	60 (40%)

**Table 2 diagnostics-14-00072-t002:** Descriptive analysis of characteristics at last postpartum visit (3 months) *n* (%).

Characteristic	POC VL, N = 77	SOC VL, N = 74	Overall, N = 151
Birth outcome of this pregnancy			
Liveborn	64 (91%)	61 (95%)	125 (93%)
Stillborn/neonatal death/abortion	6 (8.6%)	3 (4.7%)	9 (6.7%)
Mode of delivery			
Vaginal birth	68 (92%)	55 (85%)	123 (88%)
Planned c-section	2 (2.7%)	5 (7.7%)	7 (5.0%)
Emergency c-section	4 (5.4%)	5 (7.7%)	9 (6.5%)
Had separated with partner by postpartum visit	4 (5.9%)	8 (12%)	12 (9.1%)
On ART at postpartum visit	75 (99%)	72 (97%)	147 (98%)
Self-reported ART adherence at visit 3 months postpartum			
Excellent	40 (60%)	47 (73%)	87 (66%)
Good/very good	24 (36%)	16 (25%)	40 (31%)
Fair/poor	3 (4.5%)	1 (1.6%)	4 (3.1%)
Undetectable viral load (≤50 copies) at postpartum visit among participants with follow-up viral load data (*n* = 74)	38/43 (88%)	25/31 (81%)	63/74 (85%)
POC VL results at the time of delivery (a subset of the POC VL arm only) (*n* = 29)			
Undetectable viral load (≤50 copies) at delivery for mothers	28/29 (96.6%)	-	-
Infant tested HIV negative at delivery	29/29 (100%)	-	-

## Data Availability

Quantitative and qualitative data may be made available by authors C.C. and N.W. upon reasonable request.

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
