# Peer review of "It Soothes Your Heart”: A Multimethod Study Exploring Acceptability of Point-of-Care Viral Load Testing among Ugandan Pregnant and Postpartum Women Living with HIV"

_diagnostics, 2023, doi:10.3390/diagnostics14010072_

Round 1

Reviewer 1 Report

Comments and Suggestions for Authors

The authors showcase the outcomes of a study conducted among Ugandan pregnant women living with HIV (PWLHIV). In the study, they assessed the acceptability of a point-of-care viral load test (POC VL) and juxtaposed its efficacy in achieving viral suppression through ART adherence with that of the standard-of-care (SOC) lab-based testing approach. Results showed that there was no significant difference between the two groups in the proportion of women who were virally suppressed. In terms of medication adherence, 60% (40/77) women in the POC VL group self-reported their 30-day ART adherence as ‘excellent’ compared to 73% (47/74) women in the SOC group. The study also included a qualitative assessment, in which the authors found that the POC VL test was well-received by the women. They found it beneficial in supporting their ART adherence, in safeguarding their babies against the risk of vertical HIV transmission, and overall emotional well-being.

The paper is well written with clearly defined scope, but the study design requires further clarification. Additionally, there are concerns related to the data and conclusions as detailed below that would need to be addressed to strengthen the manuscript:

Major concerns:

  1. It is not clear why despite the same-day VL testing and adherence counseling offered to the PWLHIV in the POC VL group, only 60% indicated that their ART adherence was ‘excellent’. This seems worse than the SOC group where 73% reported excellent adherence. Authors should explore why this is the case by potentially interviewing the women to understand why despite the high acceptability and perceived benefits of ART adherence, their self-reported scores were low. Currently, this data point seems to contradict one of the study’s overall conclusions.
  2. The frequency of testing is a little unclear – lines 98, and 99 seem to indicate that SOC VL testing was done once every 12 months, but lines 101, and 102 indicate that all subjects were followed every 3 months. A study-design flowchart can be added explaining the design, group composition, and details of each visit for additional clarity. Example papers that have done so are listed here:
    1. Wuensch, Alexander, et al. "Effect of individualized communication skills training on physicians’ discussion of clinical trials in oncology: results from a randomized controlled trial." BMC cancer 17.1 (2017): 1-9.,
    2. Nayak, Samiksha, et al. "Integrating user behavior with engineering design of point-of-care diagnostic devices: theoretical framework and empirical findings." Lab on a Chip 19.13 (2019): 2241-2255.
  3. The authors could include and comment on the self-reported ART adherence scores and the VL levels measured for the POC VL and SOC VL groups at each visit and not just at 3 months post-partum. This could shed light on the perceived importance of VL levels closer to childbirth.

Minor concerns: 

  1. The authors should clarify what the selection criteria were for choosing the 22 women in the POC VL to take part in the qualitative assessment
  2. Authors should add a note on how the adherence levels were defined (i.e., excellent corresponds to 0 missed doses, etc.,). It is not clear in the current version of the manuscript how the scale was communicated to the study participants and whether it was consistent across participants and over the duration of the study.
  3. Authors should explicitly call out whether there was a difference in viral suppression and ART adherence based on which intervention group the male partners were assigned to.
  4. The percentages listed under section 3.2 on the proportion of virally suppressed women (lines 158, 159) are different from Table 2 (89% vs. 88% and 82% vs. 81%)

Author Response

November 9, 2023

Manuscript ID: diagnostics-2689263

Dear Diagnostics Editors,

Thank you very much for the opportunity to revise and resubmit our manuscript, It soothes your heart”: A multi-method study exploring acceptability of point-of-care viral load testing among Ugandan pregnant and postpartum women living with HIVfor Diagnostics’ Special Issue on Point-of-Care Diagnostics in Resource-Limited Settings. We have revised our manuscript based on the comments made by the editorial team and reviewers, which were very helpful. In this re-submission, we provide point-by-point responses to these reviewers’ comments, as well as marked and clean copies of the revised manuscript.

We very much appreciate your review and look forward to hearing from you,

The Kingasa Research Team

Author responses to reviewer comments

Reviewer 1

The authors showcase the outcomes of a study conducted among Ugandan pregnant women living with HIV (PWLHIV). In the study, they assessed the acceptability of a point-of-care viral load test (POC VL) and juxtaposed its efficacy in achieving viral suppression through ART adherence with that of the standard-of-care (SOC) lab-based testing approach. Results showed that there was no significant difference between the two groups in the proportion of women who were virally suppressed. In terms of medication adherence, 60% (40/77) women in the POC VL group self-reported their 30-day ART adherence as ‘excellent’ compared to 73% (47/74) women in the SOC group. The study also included a qualitative assessment, in which the authors found that the POC VL test was well-received by the women. They found it beneficial in supporting their ART adherence, in safeguarding their babies against the risk of vertical HIV transmission, and overall emotional well-being.

The paper is well written with clearly defined scope, but the study design requires further clarification. Additionally, there are concerns related to the data and conclusions as detailed below that would need to be addressed to strengthen the manuscript:

Major concerns:

  1. It is not clear why despite the same-day VL testing and adherence counselingoffered to the PWLHIV in the POC VL group, only 60% indicated that their ART adherence was ‘excellent’. This seems worse than the SOC group where 73% reported excellent adherence. Authors should explore why this is the case by potentially interviewing the women to understand why despite the high acceptability and perceived benefits of ART adherence, their self-reported scores were low. Currently, this data point seems to contradict one of the study’s overall conclusions.

Thank you for this excellent point. We have added the following paragraph to the Discussion, starting at Line 429: “Qualitatively, mothers described the many perceived ways in which POC VL testing improved their ART adherence. However, quantitatively, a lower proportion of mothers in the POC VL arm self-reported “excellent” ART adherence at post-partum visits compared to the mothers in the SOC VL arm (60% vs. 73%, Table 2). This discrepancy between the quantitative and qualitative data may be partly explained by mothers in the POC VL arm refraining from overselling or overestimating their ART adherence to the healthcare worker, given that due to POC VL testing, their VL and therefore their adherence, would be known and discussed at the same clinical encounter. This would not be as great of a concern for mothers in the SOC VL testing arm, who would only receive their VL results months at their next visit. Mothers in the POC VL arm often described how important it was that they made their healthcare workers proud of them based on their achievement of viral suppression. Additional hypotheses for reasons behind this discrepancy should be explored further in future studies.”

  1. The frequency of testing is a little unclear – lines 98, and 99 seem to indicate that SOC VL testing was done once every 12 months, but lines 101, and 102 indicate that allsubjects were followed every 3 months. A study-design flowchart can be added explaining the design, group composition, and details of each visit for additional clarity. Example papers that have done so are listed here:
    1. Wuensch, Alexander, et al. "Effect of individualized communication skills training on physicians’ discussion of clinical trials in oncology: results from a randomized controlled trial." BMC cancer1 (2017): 1-9., 
    2. Nayak, Samiksha, et al. "Integrating user behavior with engineering design of point-of-care diagnostic devices: theoretical framework and empirical findings." Lab on a Chip13 (2019): 2241-2255.

Thank you for this recommendation, accompanied by examples. We have added a study-design flowchart to clarify study visits and activities based on randomization arm (new Figure 1 in Methods).

  1. The authors could include and comment on the self-reported ART adherence scores and the VL levels measured for the POC VL and SOC VL groups at each visit and not just at 3 months post-partum. This could shed light on the perceived importance of VL levels closer to childbirth.

Thank you for this important point. We have now mentioned in the Limitations section that we could not assess changes in ART adherence over time in this pilot study. Tables 1 and 2 already present VL levels at baseline and post-partum (POC group: 86% undetectable at baseline and 88% at post-partum; SOC group: 80% undetectable at baseline and 81% at post-partum). The small sample size and high proportions of viral suppression limit our power to detect any subtle differences in viral suppression over time and by group in this pilot study.

POC VL testing was only conducted at delivery among a small subset of mothers in the POC VL arm (n=29), as it was a study procedure piloted in this study and was not always possible to obtain for women who delivered at night or on weekends. We have now added a row to Table 2 presenting the POC VL results at delivery for these 29 mothers and their infants. These data are contextualized and supported by the qualitative findings, as all women in the qualitative sample had received POC VL testing at the time of delivery or soon thereafter. 

Minor concerns:  

  1. The authors should clarify what the selection criteria were for choosing the 22 women in the POC VL to take part in the qualitative assessment.

The procedures for selecting the 22 women in the POC VL arm are now described further in the Methods, starting from Line 141: “A purposively sampled subset of PWLHIV were invited to participate in a single in-depth qualitative interview; only PWLHIV who were randomized to the POC VL arm and had already received POC VL testing were recruited. The selection criteria for the qualitative sub-sample were PWLHIV who: (a) had received POC VL at the time of delivery, (b) had not yet completed their exit visit at 3 months post-partum, and (c) were reachable by telephone for scheduling.”

  1. Authors should add a note on how the adherence levels were defined (i.e., ‘excellent’ corresponds to 0 missed doses, etc.,). It is not clear in the current version of the manuscript how the scale was communicated to the study participants and whether it was consistent across participants and over the duration of the study.

The definition of self-reported adherence levels is presented on Line 140: “All participants were asked at their 3-month post-partum visit to self-report their perceived ART adherence in the past 30 days using a 5-point Likert scale (ranging from excellent, very good, good, fair, poor, to very poor).”

  1. Authors should explicitly call out whether there wasa difference in viral suppression and ART adherence based on which intervention group the male partners were assigned to.

We first added the following text for additional information in the Methods, starting at Line 93: “For the present analysis focused on comparing POC VL to SOC VL testing, we grouped all mothers who were randomized to POC VL testing vs. SOC VL testing, regardless of the male partner intervention arm to which they were also randomized. All multivariate analyses assessing differences in outcomes by VL testing method were adjusted for male partner intervention randomization arm.”

We have also added the following results to the end of Paragraph 3.2 (Line 199): “The randomization for male partner testing interventions was not associated with differences in post-partum viral suppression or self-reported ART adherence (data not shown).”

  1. The percentages listed under section 3.2 on the proportion of virally suppressed women (lines 158, 159) are different from Table 2 (89% vs. 88% and 82% vs. 81%)

Thank you for identifying this discrepancy; we have fixed the text in Line 176 to match the data presented in Table 2 (correct number: 88% vs. 81%).

Reviewer 2 Report

Comments and Suggestions for Authors

The authors provided great insight into the acceptability of POC viral loading testing amongst pregnant and postpartum women living with HIV. The methods used to assess acceptability are sound. 

Couple of major issues:

1. There seems to be a lack of quantitative data reporting if the POCVL increased adherence. How did the participants rate their adherence before enrollment in the study and how did the POCVL adherence group compare to the SOCVL adherence group? Since increased adherence is one major outcome of the study, this data would be useful to report.

2. Part of the method states that different arms were given XVL+invitation to the male partners for wellness or testing. However, there was no discussion about the outcome of the male participation. Maybe remove this part of the method or include the data/discussion?

3. The majority of the discussion focused on the data collected from the POCVL group but very little comparing info from the SOCVL group to assess if POCVL would be more effective in maintaining low viral load.

4. How did adherence change during pregnancy vs postpartum for the two groups?

5. Did POCVL improve viral load outcomes? Since all participants were on ART the majority had low VL load to begin with. Is there any data suggesting that POCVL improved low VL maintenance over SOCVL?

6. One concluding statement is that improved infant acquiring HIV. However, on page 5 it was mentioned that all babies were HIV negative. This suggest that POCVL performed on the same level as SOCVL and did not necessarily improved the outcome. 

The impact on mental wellness is clearly evident as a positive outcome of POCVL testing. The addition of data to reflect improved adherence compared to SOCVL during pregnancy and postpartum will be helpful to boost the significant impact of the research.

Author Response

November 9, 2023

Manuscript ID: diagnostics-2689263

Dear Diagnostics Editors,

Thank you very much for the opportunity to revise and resubmit our manuscript, It soothes your heart”: A multi-method study exploring acceptability of point-of-care viral load testing among Ugandan pregnant and postpartum women living with HIVfor Diagnostics’ Special Issue on Point-of-Care Diagnostics in Resource-Limited Settings. We have revised our manuscript based on the comments made by the editorial team and reviewers, which were very helpful. In this re-submission, we provide point-by-point responses to these reviewers’ comments, as well as marked and clean copies of the revised manuscript.

We very much appreciate your review and look forward to hearing from you,

The Kingasa Research Team

Reviewer 2

The authors provided great insight into the acceptability of POC viral loading testing amongst pregnant and postpartum women living with HIV. The methods used to assess acceptability are sound. 

Couple of major issues:

  1. There seems to be a lack of quantitative data reporting if the POCVL increased adherence. How did the participants rate their adherence before enrollment in the study and how did the POCVL adherence group compare to the SOCVL adherence group? Since increased adherence is one major outcome of the study, this data would be useful to report.

Thank you for this important comment. We have now added baseline self-reported ART adherence to Table 1.However, self-reported ART adherence data at baseline and 3-months post-partum were not ascertained using the same question and cannot be accurately compared. At the post-partum visit, self-reported ART adherence was assessed using a Likert scale ranging from excellent to poor (described in Methods); however, at baseline, self-reported ART adherence was assessed using the number of days lapsed since the last ART dose. These data have now been added to Table 1.  We have added this as a limitation to the discussion section, Line 467: “One outcome of this analysis, self-reported ART adherence at the post-partum visit measured using a Likert scale, was not collected at baseline, limiting our ability to assess changes in ART adherence between the time of baseline and post-partum by randomization arm.”

  1. Part of the method states that different arms were given XVL+invitation to the male partners for wellness or testing. However, there was no discussion about the outcome of the male participation. Maybe remove this part of the method or include the data/discussion?

While we had mentioned in the Methods section (Line 154) that this present analysis only includes data for women, we recognize that further clarification on the randomization arms for male partner interventions is needed for the reader. Please see our response to Reviewer 1’s comment 3 under “minor concerns” regarding the male partner arms.

  1. The majority of the discussion focused on the data collected from the POCVL group but very little comparing info from the SOCVL group to assess if POCVL would be more effective in maintaining low viral load.

Thank you for this important comment.  We have expanded on paragraph 2 of the Discussion section: “Though our quantitative analysis did not find a statistically significant difference in women’s achievement of viral suppression by VL testing type, this pilot study was limited by a small sample size and limited statistical power, given that >80% of PWLHIV were already virally suppressed at study enrollment. A larger study involving a greater proportion of virally unsuppressed PWLHIV at baseline is needed to assess differences in treatment outcomes by VL testing type over time. The qualitative findings contributed important insights into the reasons why women believed that POC VL testing directly improved their motivation to adhere to ART compared to SOC VL testing and is likely still an important factor, even if not reflected statistically. This demonstrates the value of a multi-method approach, as the qualitative analysis provided greater context beyond the quantitative findings alone.”

  1. How did adherence change during pregnancy vs postpartum for the two groups?

Thank you for this comment. We have added the following as a limitation in the Discussion, Line 471: “One outcome of this analysis, self-reported ART adherence at the post-partum visit, was not collected at baseline, limiting our ability to assess changes in ART adherence between the time of baseline and the 3-month post-partum visit by randomization arm. Given the short follow-up time and small sample size for this pilot RCT, we could not assess changes in ART adherence over time.”

  1. Did POCVL improve viral load outcomes? Since all participants were on ART the majority had low VL load to begin with. Is there any data suggesting that POCVL improved low VL maintenance over SOCVL?

We agree with the reviewer's concern about assessing differences in VL outcomes by randomization arm, given that most PWLHIV were virally suppressed at baseline. Please refer to our response to comment 3 above and the expanded paragraph 2 in the Discussion.

  1. One concluding statement is that improved infant acquiring HIV. However, on page 5 it was mentioned that all babies were HIV negative. This suggest that POCVL performed on the same level as SOCVL and did not necessarily improved the outcome. 

Thank you for the opportunity to clarify this important concluding point.  We believe the reviewer is referring to our concluding sentence on Line 487: “POC VL testing was viewed as an important tool that women believed improved their ART adherence, gave them the information necessary to protect their children from vertical HIV acquisition at different stages prenatally and post-partum, and improved their emotional wellbeing.” We have rephrased this sentence now to read: “Qualitative findings indicated that women viewed POC VL testing as an important tool that improved their ART adherence, gave them information and motivation to protect their children from vertical HIV acquisition during pregnancy and post-partum, and improved their emotional wellbeing. This concluding sentence referred to qualitative data on women’s experiences with POC VL testing and was not based on a quantitative comparison of infant HIV acquisition between the POC VL vs. SOC VL arms. Although we were unable to assess this outcome quantitatively, qualitative data strongly suggested that women believed that knowing their VL results immediately gave them the information and tools necessary to make informed choices that protected their babies from acquiring HIV pre- and post-delivery (e.g., during breastfeeding). Please notify us if this comment concerns a different sentence needing further clarification or rewording; we would happily rework it.

  1. The impact on mental wellness is clearly evident as a positive outcome of POCVL testing. The addition of data to reflect improved adherence compared to SOCVL during pregnancy and postpartum will be helpful to boost the significant impact of the research.

We agree with the reviewer that the qualitative data suggest that POC VL had a positive impact on mothers’ emotional wellbeing; however, we did not assess mental health quantitatively. A larger study would be needed to quantitatively assess associations between POC VL testing and improved mental health outcomes (e.g., depression and anxiety measures). Despite qualitative evidence that POC VL improved and supported women’s post-partum ART adherence, the quantitative data did not show any statistical improvement in self-reported adherence in the POC VL group compared to the SOC VL group. Please refer to our response to Reviewer 1 comment 1 under “major concerns.”

Reviewer 3 Report

Comments and Suggestions for Authors

The manuscript is a simple translational study demonstrating the value of point-of-care diagnostics for improving the motivation of medication adherence for the HIV population.  While uncomplicated and not persay novel, this is a critical study to provide evidence that can improve health outcomes, and is thus important and noteworthy.

Notably, the study has greater implications for the medical and diagnostics community, providing a key study demonstrating the power of companion diagnostics and point-of-care testing to health outcomes and patient motivations.

A couple of things could and/or need to be addressed to further improve the quality of the paper.

1)  Table 2 should have its last point moved up or made clearer that it doesn't belong with Self-Reported ART adherance.  It looks in the table as its a sub-set of that while in reality it is something completely different.

2)  The discussion needs to be reorganized to improve focus.  Notably, there is a talk about location being a phenomena and an indicative factor, but it seems like its included in the middle as a stray thought...perhaps the use of sub-titles could help focus and transition between major key points and the note about location could be moved towards the end to talk about the greater impact of the study and its utility?  It seems odd to throw this line into a larger discussion of the testing results without breaking down the survey's by region and even bringing up rural POC.  Plus, even in urban environments, POC diagnostics appear to have significant benefit.

3)  The actual POC test used is buried in the methods section with little details.  Additionally, there is little discussion as to infrastructure needs (reader, etc.) and cost to execute these assays, especially compared to the standard clinical assays.  This NEEDS to be addressed both in the methods session and the discussion to discuss the overall impacts of POC diagnostics, or key steps needed to further implement these POC diagnostics in low-income, austere locations.  

Author Response

November 9, 2023

Manuscript ID: diagnostics-2689263

Dear Diagnostics Editors,

Thank you very much for the opportunity to revise and resubmit our manuscript, It soothes your heart”: A multi-method study exploring acceptability of point-of-care viral load testing among Ugandan pregnant and postpartum women living with HIVfor Diagnostics’ Special Issue on Point-of-Care Diagnostics in Resource-Limited Settings. We have revised our manuscript based on the comments made by the editorial team and reviewers, which were very helpful. In this re-submission, we provide point-by-point responses to these reviewers’ comments, as well as marked and clean copies of the revised manuscript.

We very much appreciate your review and look forward to hearing from you,

The Kingasa Research Team

Reviewer 3

The manuscript is a simple translational study demonstrating the value of point-of-care diagnostics for improving the motivation of medication adherence for the HIV population.  While uncomplicated and not persay novel, this is a critical study to provide evidence that can improve health outcomes, and is thus important and noteworthy. Notably, the study has greater implications for the medical and diagnostics community, providing a key study demonstrating the power of companion diagnostics and point-of-care testing to health outcomes and patient motivations. A couple of things could and/or need to be addressed to further improve the quality of the paper.

  1. Table 2 should have its last point moved up or made clearer that it doesn't belong with Self-Reported ART adherence.  It looks in the table as its a sub-set of that while in reality it is something completely different.

We have now rearranged the variables in Table 2 to avoid confusion for readers.

  1. The discussion needs to be reorganized to improve focus.  Notably, there is a talk about location being a phenomena and an indicative factor, but it seems like its included in the middle as a stray thought...perhaps the use of sub-titles could help focus and transition between major key points and the note about location could be moved towards the end to talk about the greater impact of the study and its utility?  It seems odd to throw this line into a larger discussion of the testing results without breaking down the survey's by region and even bringing up rural POC.  Plus, even in urban environments, POC diagnostics appear to have significant benefit.

Thank you for pointing out the tangential sentences on the location of POC VL delivery. We have removed these sentences from the paper as they were irrelevant to the discussion.

  1. The actual POC test used is buried in the methods section with little details.  Additionally, there is little discussion as to infrastructure needs (reader, etc.) and the cost to execute these assays, especially compared to the standard clinical assays.  This NEEDS to be addressed both in the methods session and the discussion to discuss the overall impacts of POC diagnostics or key steps needed to further implement these POC diagnostics in low-income, austere locations.  

We agree with this reviewer that our paper would benefit from additional details on the POC VL platform. We have added the following paragraph to the Methods, starting at Line 105: “POC VL testing was performed using the GeneXpert® HIV-1 Viral Load test (Cepheid, Sunnyvale, CA),17 which has been tested and validated for use in sub-Saharan African settings with a sensitivity and specificity of 94% and 99%, respectively.18 This self-contained, rapid polymerase chain reaction (PCR) test detects the number of HIV copies per milliliter of plasma in an average time of 90 minutes. The POC VL machine used in this study could process up to four samples simultaneously, allowing for larger testing volumes for optimized efficiency. At study visits with VL testing for women in the POC VL arm, research nurses collected a blood sample at the beginning of the clinical encounter so that the laboratory technician could process it. At the same time, the participant completed their study visit. As soon as the laboratory technician processed the POC VL results, the research nurse delivered the results to the mother, accompanied by personalized VL and adherence counseling. When samples were collected late in the afternoon close to laboratory closure, results and counseling were provided to mothers in the POC VL arm the following day via telephone.”

We added the following paragraph to the Discussion section, starting at Line 448: “As HIV treatment initiation and monitoring guidelines recommend less CD4 testing, the demand for decentralized VL testing as the preferred method for HIV treatment monitoring is growing substantially across sub-Saharan Africa.22 With centralized laboratories overburdened by processing needs for HIV, TB, COVID-19, and other diseases, POC VL testing is the preferred method for scale-up in high-burden HIV settings thanks to reduced turn-around times. Reassuringly, POC VL testing in sub-Saharan Africa has shown that it can be accurate and feasibly implemented at reasonable costs.23-25 For example, the average costs of a POC VL test and a SOC VL test using centralized laboratory testing were $33.71 and $28.62, respectively, in Malawi, and $24.25 per POC VL test and $25.65 per SOC VL test in Kenya.23,24 POC VL testing implementation would require a dedicated and trained POC technician, which may require task shifting among staff. POC VL testing can be accurately conducted by trained nonlaboratory, lay worker staff, which may be necessary in more rural, resource-limited clinics or communities.26 Innovative and cross-cutting solutions are needed to address common health systems challenges with POC VL testing, including equipment procurement and maintenance, supply chain management for cartridges, and limited staff availability for after-hours/weekend processing.25

Round 2

Reviewer 1 Report

Comments and Suggestions for Authors

The authors have satisfactorily addressed most of the previously raised concerns. The manuscript is nearly ready for publication, pending a small update as outlined below.

The addition of a flowchart and further explanations has clarified the study design. However, there are still details that need to be included for completeness. Specifically, the authors noted that the POCVL group was followed every three months, but VL testing did not occur at each visit. The flowchart should be revised to reflect this, clearly indicating the procedures performed at each study visit, including whether patients received counseling or VL testing. In the case of the SOCVL cohort, their initial VL test was administered during a prenatal visit, with a follow-up test three months postpartum. The manuscript currently lacks clarity on when and how the study participants were informed of their test results and counseled, especially considering their next scheduled visit was 12 months later. The revised flowchart should also address this, providing detailed information on the timing and method of communicating test results to participants. 

Author Response

We thank Reviewer 1 for the excellent comments. Please find our responses below: 

The authors have satisfactorily addressed most of the previously raised concerns. The manuscript is nearly ready for publication, pending a small update as outlined below.

The addition of a flowchart and further explanations has clarified the study design. However, there are still details that need to be included for completeness. Specifically, the authors noted that the POCVL group was followed every three months, but VL testing did not occur at each visit. The flowchart should be revised to reflect this, clearly indicating the procedures performed at each study visit, including whether patients received counseling or VL testing.

We agree with the reviewer that it is important to indicate in the flowchart which visits participants received VL testing. We have added more details to the figure, as well as * to the specific visits during which VL testing occurred

In the case of the SOCVL cohort, their initial VL test was administered during a prenatal visit, with a follow-up test three months postpartum. The manuscript currently lacks clarity on when and how the study participants were informed of their test results and counseled, especially considering their next scheduled visit was 12 months later. The revised flowchart should also address this, providing detailed information on the timing and method of communicating test results to participants. 

Thank you very much for the opportunity to revise this section. We have added more details to the study procedures, specifically for the SOC VL arm. We have also added the following sentence to the Methods on Line 104: For this study, the SOC VL group provided their blood samples for two time-points of viral load testing: 1) at their enrollment visit during pregnancy, and received their results and counseling 3 months later at their next study visit, and 2) at their 3-month post-partum visit, and received their results at the clinic 3 months later (6-month post-partumFigure 1)."